# Do Food and Nutrition Policies in Ethiopia Support the Prevention of Non-Communicable Diseases through Population-Level Salt Reduction Measures? A Policy Content Analysis

**DOI:** 10.3390/nu15071745

**Published:** 2023-04-03

**Authors:** Dejen Yemane Tekle, Emalie Rosewarne, Joseph Alvin Santos, Kathy Trieu, Kent Buse, Aliyah Palu, Anne Marie Thow, Stephen Jan, Jacqui Webster

**Affiliations:** 1The George Institute for Global Health, The University of New South Wales, Sydney, NSW 2042, Australia; 2School of Public Health, Mekelle University, Mekelle 1871, Ethiopia; 3The George Institute for Global Health, Imperial College London, London NW9 7PA, UK; 4Menzies Centre for Health Policy, School of Public Health, University of Sydney, Sydney, NSW 2006, Australia

**Keywords:** policy analysis, documentary analysis, salt reduction, noncommunicable diseases, policy cube

## Abstract

Introduction: Despite the importance of salt reduction to health outcomes, relevant policy adoption in Ethiopia has been slow, and dietary consumption of sodium remains relatively high. Aim: This analysis aims to understand the content and context of existing food-related policy, strategy, and guideline documents to identify gaps and potential opportunities for salt reduction in Ethiopia in the wider context of global evidence-informed best practice nutrition policy. Methods: Policy documents relevant to food and noncommunicable diseases (NCDs), published between 2010 and December 2021, were identified through searches of government websites supplemented with experts’ advice. Documentary analysis was conducted drawing on the ‘policy cube’ which incorporates three dimensions: (i) comprehensiveness of policy measures, which for this study included the extent to which the policy addressed the food-related WHO “Best Buys” for the prevention of NCDs; (ii) policy salience and implementation potential; and (iii) equity (including gender) and human rights orientation. Results: Thirty-two policy documents were retrieved from government ministries, of which 18 were deemed eligible for inclusion. A quarter of these documents address diet-related “Best Buys” through the promotion of healthy nutrition and decreasing consumption of excess sodium, sugar, saturated fat, and trans-fats. The remainder focuses on maternal and child health and micronutrient deficiencies. All documents lack detail relating to budget, monitoring and evaluation, equity, and rights. Conclusions: This review demonstrates that the Government of Ethiopia has established policy frameworks highlighting its intention to address NCDs, but that there is an opportunity to strengthen these frameworks to improve the implementation of salt reduction programs. This includes a more holistic approach, enhanced clarification of implementation responsibilities, stipulation of budgetary allocations, and promoting a greater focus on inequities in exposure to nutrition interventions across population groups. While the analysis has identified gaps in the policy frameworks, further qualitative research is needed to understand why these gaps exist and to identify ways to fill these gaps.

## 1. Introduction

Noncommunicable diseases (NCDs) are the leading cause of premature deaths, responsible for 41 million deaths each year, equivalent to 71% of all premature deaths globally [1]. NCDs disproportionately affect low- and middle-income countries (LMICs), with three-quarters (32 million deaths) of NCD-related premature deaths occurring in LMIC populations, where they impose large, avoidable burdens on health care systems as well as human, social, and economic costs [2]. In sub-Saharan Africa (SSA), NCDs cause a large and growing burden of death and illness, with estimates of regional prevalence of hypertension at 48%, obesity at 20% and diabetes at 5% [3]. In SSA, it is projected that NCDs will overtake infectious diseases as the leading cause of morbidity and mortality by 2030 [4]. In Ethiopia, NCDs account for four in ten deaths (42%), of which 27% are premature deaths before 70 years of age. Disability Adjusted Life Years (DALYs) from NCDs increased from below 20% of the total burden in Ethiopia in 1990 to 69% in 2015 [5]. Without action, the World Health Organization (WHO) projects this burden to increase to 7 in 10 premature deaths by 2040 [6,7].

NCDs are mainly attributable to four modifiable risk factors: diet, alcohol, tobacco, and physical inactivity. The contribution of poor diet to the burden of NCDs exceeds the combined contribution from alcohol, tobacco, and physical inactivity [8]. Poor diets mainly reflect a predominantly unhealthy global food environment, dominated by processed (and ultra-processed) foods high in sugar, saturated fat, artificial trans-fatty acids, and sodium [8].

A recent Global Burden of Disease study showed that a diet high in sodium (consumed mostly in the form of salt) is ranked as the leading dietary risk factor causing deaths and Disability Adjusted Life Years (DALYs) globally [9]. Diets high in sodium are responsible for over 1.8 million NCD deaths and 45 million Disability Adjusted Life Years (DALYs) [10]. There is clear evidence that diets high in sodium contribute to high blood pressure [11,12,13], which alone accounted for an estimated 10.4 million deaths and 218 million DALYs in 2017 [14]. High sodium intakes and high blood pressure increase the risk of cardiovascular diseases [15,16], gastric cancer [17], and chronic kidney disease [18,19].

Citing the overwhelming evidence linking high salt intake to NCDs, WHO has identified four interventions to lower population salt intake as “Best Buys” for the prevention and control of diet-related NCDs. These include behavior change communication, the reformulation of food products to contain less salt, the implementation of front-of-pack labeling and the provision of lower salt options in public institutions (e.g., schools and workplaces) [20,21]. In 2013, WHO set a global target of a 30% relative reduction in population salt intake to be achieved by 2025 and an absolute level of no more than 5 g/day, which all member states signed up to achieve [20,22]. Despite the Ethiopian government signing up to this target in the WHO Global Action Plan to reduce the burden of NCDs [23], there seems to be limited action to reduce sodium consumption in Ethiopia. Average salt intake is estimated to be around 8.3 g per day [24], with 96.2% of the population exceeding the 5 g/day limit. The majority of salt intake is from salt added during cooking as well as from home-prepared condiments and spices such as berbere (spice mix), mitmita (seasoning) and shiro (chickpea spread) [25]. Although salt intake from processed foods is increasing.

As part of a broader policy analysis to understand barriers and enablers to salt reduction programs in Ethiopia, this documentary analysis aims to understand the content and context of existing food-related guidelines, policies, and strategic documents. This is with a view to identifying gaps and understanding where and how policies could be strengthened to effectively address the growing burden of NCDs through salt reduction in the wider context of global best practice nutrition policies. Together with the broader policy analysis, the documentary analysis will be used to help the country make informed decisions on the formulation, adoption, planning and implementation of programs to reduce population salt intake in Ethiopia.

## 2. Materials and Methods

### 2.1. Research Setting

The study was conducted in Ethiopia, the second-most populous country in SSA. Ethiopia is a federal state comprising nine regions and two city administrative councils (Dire Dawa and Addis Ababa). Administrative power tiers start from the federal government, then moves to the regions, then zones, woreda (districts), and finally kebele (the smallest government administrative units in the country). The Federal Government has the mandate to develop and enact health policies to be adopted by the regional states to address societal problems.

### 2.2. Search Strategy

Our focus was on public policy, including legislation and policy documents published by government bodies. We aimed to identify policies relevant to diet-related NCD prevention, including policies that impact the food supply as well as diets. We searched governmental websites (Ministry of Health, Ministry of Trade and Industry, Ministry of Agriculture, Ethiopian Public Health Institute, and the Ethiopian Food and Drug Administration) from 2010 to December 2021 and supplemented this information with suggestions made by contacts the researchers have with staff in the Ministry of Health and the WHO country office. We also consulted with local nutrition experts to verify that we had identified all relevant policy documents. Both Amharic and English search terms were used to identify legislation and policy documents. Documents in Amharic were then translated into English for the purpose of the readers. We used two groups of search terms: (1) policies OR proclamations OR programs OR national action plans OR guidelines OR strategies AND (2) health OR NCDs OR noncommunicable diseases OR Nutrition OR food. Records were managed by EndNote X9.0 software.

### 2.3. Inclusion and Exclusion Criteria

National policies, proclamations, programs, national action plans, guidelines, and strategies containing information relevant to food and nutrition in the context of NCDs and published between 2010 and December 2021 were included. Documents that had been superseded were excluded.

### 2.4. Data Extraction and Analysis

This documentary policy analysis reviewed the content of national policies inside and outside the health sector using a “policy cube” that brings together three axes to understand the strength of a policy framework to combat diet-related NCDs [21] (See Figure 1). The axes are: (1) *comprehensiveness of policy measures*; for this analysis, we interpreted this to include the extent to which the policy documents are aligned to tackle NCD prevention, address food and nutrition and/or reflect the WHO ‘Best Buy’ recommendations relating to salt; (2) *policy salience and implementation potential*: for this study, this included the explicit mention or description of monitoring and evaluation mechanisms, associated budgets and the level of policy authority and systems of accountability in the policy documents; and (3) *equity, gender and rights orientation:* which is the extent to which the documents reference and reflect the principles of equity (including gender) and human rights. We adopted the policy cube as it provides a structured approach to examining the strength of existing dietary policies and identifying areas where additional focus is needed. We reviewed each policy for any language on each axis and collated this information for an overall assessment of the policy environment.

For the purpose of this study, ‘policy authority’ was interpreted as the legitimacy to influence and is related to the status of the government body issuing the policy. Hence, we categorized relevant documents in relation to their relative ‘authoritativeness’ as: high (document that has been subject to national consultation, gone through a parliamentary committee, and been legislated by two houses of parliament); moderate (documents subject to a relatively inclusive consultation process and rubber-stamped by a ministerial technical committee but not legislated by any of the houses of parliament); or low (documents based on international best-practice and implementation plans). The treatment of equity was assessed in relation to whether the policy document acknowledged that some groups in the population are more likely to be differentially exposed to NCDs in general and higher salt intake in particular due to their employment, gender, disability, age, geography and social status, and whether efforts were outlined in relation to how to reach and/or target those groups. The presence or absence of reference to rights-based approaches was also assessed based on the articulation/mention of principles of human rights in the NCD policies.

An Excel data extraction template was developed based on the ‘policy cube’ to collect the following information: title; policy type; time frame; comprehensiveness of policy measures; policy salience and implementation potential; equity (including gender) and human rights orientation. Relevant information was extracted and analyzed narratively in relation to key themes from the ‘policy cube’ by one researcher, D.Y.T., in consultation with a second researcher, J.W., as required.

## 3. Results

### 3.1. Description of Documents Reviewed

We retrieved 33 policy documents from the search of relevant websites. After full-text screening, 15 documents were excluded because they were superseded by a more recent policy document. The 18 remaining documents were published between 2010 and 2020 and included the following types of documents: two policies, one proclamation, five programs, nine national plans or strategies and one guideline (Table 1). Fourteen out of the eighteen eligible documents were out of date but included because they had not been superseded. The National Strategic Plan for the Prevention and Control of Major Non-Communicable Diseases [27], the National Food and Nutrition Policy [21], the Second Generation Health Extension Program [25] and the National Health Policy [23] were the only current policies.

### 3.2. Comprehensiveness of Policy Measures

#### 3.2.1. Alignment with Diet-Related NCDs

Relevant documents ranged from the overarching Health Sector Development Plans to the more specific National Food and Nutrition Policy. The majority (*n* = 12) of the policy documents focused on preventing and reducing morbidity and mortality attributed to NCDs through health promotion and the adoption of healthy lifestyles, addressing risk factors and treating NCDs.

The remaining six documents (Nutrition Sensitive Agriculture Strategic Plan [26], National school health and nutrition strategy [34], Agriculture and Natural Resources Sector Growth and Transformation Plan II [35], Agricultural growth program II [36], Ethiopia Nutrition Advocacy Plan [38] and the Health sector development program IV [29]) focused on food security and/or micronutrient deficiencies (Table 2).

#### 3.2.2. Extent to Which Food and Nutrition Issues Are Addressed

All the documents addressed food and nutrition issues to varying degrees. Three documents from the Agriculture Ministry—the Nutrition Sensitive Agriculture Strategic Plan [26], the Agriculture and Natural Resources Sector Growth and Transformation Plan II [35] and the Agricultural Growth Program II [36]—promoted a food-based approach to agricultural development to improve the consumption of diversified diets through diversification of crop, fruit and livestock production. The Nutrition Sensitive Agriculture Strategic Plan recognizes diet diversification as pivotal to the attainment of food and nutrition security, but there is no mention of NCDs in any of these three plans.

More than half (*n* = 9) of the food and nutrition-related programs focused on the critical periods of the first 1000 days (pregnancy through the first two years of a child’s life) and provided a framework for coordinated implementation of nutrition interventions to ensure food safety and prevent malnutrition. Programs included optimal breastfeeding, optimal complementary feeding, mitigation and prevention of micronutrient deficiencies, water sanitation and hygiene, deworming, food fortification and management of acute malnutrition) [22,23,27,29,33,34,36,37,40] rather than preventing NCDs (Table 2).

Five documents (the National Strategic Plan for the Prevention and Control of Major Non-Communicable Diseases [27], Guidelines on Clinical and Programmatic Management of Major Non-Communicable Diseases [33], NSAP for Prevention and Control of Non-Communicable Diseases [32], The National Health Promotion and Communication Strategy [33], and Second Generation HEP [25] promoted adequate consumption of fruits, vegetables and healthy and traditionally acceptable foods and discouraged unhealthy diets. The documents encouraged consideration of policy and legislation, like subsidies on vegetables and fruits as well as increasing the tax on processed foods and drinks to decrease consumption of excess salt, sugar and sweeteners, saturated fats, trans fatty acids and/or refined carbohydrates (Table 2).

One document, the Food, Medicine and Health Care Administration and Control Proclamation [24], outlines standards for nutrition labeling, licensing of foods, regulating transregional food and the safety and quality of food (Table 2).

#### 3.2.3. Content Relevant to Salt Reduction

Five documents contained at least one reference to the WHO “Best Buys” for salt reduction. This includes behavior change communication in the Second-Generation Health Extension Program [25] and Guidelines on Clinical and Programmatic Management of Major NCDs [33]; reformulation of food products to contain less salt in the National Cancer Control Plan [32] and Second-Generation Health Extension Program [35]; implementation of front-of-pack labeling in the Guidelines on Clinical and Programmatic Management of Major NCDs [33]; and restricting marketing of foods high in salt and providing lower salt options in public institutions (e.g., schools and workplaces) in the National Strategic Action Plan (NSAP) for prevention and control of non-communicable diseases in Ethiopia [32]. Whereas the National Strategic Plan for the Prevention and Control of Major Non-Communicable Diseases [27] includes all the WHO “Best Buys” interventions related to salt reduction.

The remainder of the policy documents (*n* = 13) had no detail about salt or highlighted only salt iodization and iodine deficiency (Table 2).

### 3.3. Policy Salience and Implementation Potential

#### 3.3.1. Policy Authority

The authoritativeness of the included policy documents ranged from high (one document) to moderate (fifteen documents) to low (two documents) and is depicted in Table 3 below. Except for the Guidelines on Clinical and Programmatic Management of Major Non-Communicable Diseases [33], which was at the lower end of the authority spectrum, the rest of the four policy documents that contained elements of WHO “best buys” related to salt reduction were at the moderate end of the authority spectrum [35,40,41].

#### 3.3.2. Budget and Systems of Accountability

Except for in the National Nutrition Program II [31], the National Nutrition Program I [30], the Agriculture and Natural Resources Sector Growth and Transformation Plan II [35], the Health Sector Transformation Plan [39] and the National Strategic Plan for the Prevention and Control of Major Non-Communicable Diseases [27] there was no mention of specific dedicated budgets in any of the policy documents. Instead, there are suggestions to mobilize funding and resources from the Ethiopian Government Treasury, development partners, the private sector and community participation (Table 2).

Regarding systems of accountability, in the majority (*n* = 13) of the documents, a national multi-sectoral food and nutrition governing body is bestowed with the required authorities to govern and coordinate the implementation of the food and nutrition activities. The accountability, legal framework and functional organization structure of the governing body are responsible for facilitating the implementation of all food and nutrition activities from federal to kebele (the smallest administrative unit in Ethiopia) levels. The Federal Ministries of Education, Health and Agriculture are mandated to take the leadership role through their Nutrition Coordinating Body, under the Office of the Deputy Prime Minister, to ensure the delivery of an integrated strategy at all levels and to ensure responsibility and accountability by the different sectors for the different strategies mentioned above. Monitoring of key indicators for major NCD programs is conducted by collecting data at regular intervals, and continuous evaluation of processes and outcomes of national NCDs is in place except for the Ethiopia National Health Policy [22] and the Food, Medicine and Health Care Administration and Control Proclamation No. 661/2009 [24] (Table 2).

### 3.4. Equity, Gender and Human Rights Orientation

Only three documents (the National School Health and Nutrition strategy [34], the National Food and Nutrition Policy [21], and the National Strategic Plan for the Prevention and Control of Major Non-Communicable Diseases [32]) reflect all the principles of gender, human rights and equity. In contrast, two documents (Food, Medicine and Health Care Administration and Control Proclamation No. 661/2009 [24] and Second-Generation HEP [25] include no specific language concerning human rights, gender equality or equity. Three-quarters of the documents address only one or two of equity, gender or human rights (Table 2).

#### Summarizing the Findings in a Policy Cube

Figure 2 shows the extent to which the NCD policy framework meets the criteria assessed in our review—policy comprehensiveness, effective means of implementation and equitable rights-based approaches. As depicted in Figure 2, the policy documents have reflected moderate comprehensiveness. In contrast, the relevant documents have shown relatively low authority and do not identify accountability mechanisms or budget sources; hence, a low implementation potential is recorded.

## 4. Discussion

This documentary analysis of Ethiopian policies is the first to understand the content and context of existing guidelines, policies and strategic documents relevant to salt reduction. We adopted the NCD “Policy Cube”, a tool to assess the robustness of policy frameworks for the prevention and control of NCDs in terms of the comprehensiveness of relevant measures against a global gold standard (WHO NCDs Best Buys), government prioritization (i.e., the political salience of the policy), and orientation towards addressing equity (including gender) and human rights. This has helped to identify where and how salt reduction policies could be strengthened to address the growing burden of NCDs more effectively in the country.

This study identified 12 policy documents that highlighted the need to address diet related WHO “best buys” through promoting healthy nutrition and discouraging unhealthy diets. Five of the twelve documents [30,38,42,43] included specific actions on salt reduction and three quarters of them are at the moderate authoritativeness spectrum. This shows that the Federal Government of Ethiopia has recognized and initiated salt reduction policy responses to the growing epidemic of NCDs; however, there is opportunity for salt reduction to be incorporated into several other policy documents. Whilst several of the policies referred to the need to improve food and nutrition for NCD prevention and mentioned one or more of the WHO’s best buys for salt reduction, there were no clear mechanisms for implementation. Given the established health benefits of dietary salt reduction [42,44], there is a need to increase the priority on the political agenda and convince government leadership, including ministries beyond the health ministry, to take specific actions like establishing policies to reduce salt intake through front-of-pack nutrition labeling systems and food reformulation strategies, supporting the food and beverage industries to cut sodium levels in processed foods, committing to adequate budget allocation and the establishment of mechanisms for monitoring and accountability to address the growing burden of diet-related NCDs in the country [21,45]. Without adequate attention, such gaps are liable to feed into widening health problems and overstretch Ethiopia’s already fragile public health systems.

While not outlined in the policy documents, the Ethiopian government has recently launched a national salt reduction mass media campaign as part of the effort to increase awareness about the dangers of a high-salt diet [43]. This may reflect findings elsewhere regarding the appeal of behavioral approaches and policies for governments [46], in contrast to addressing the social, structural and commercial environments through programs like reformulation or reducing salt in settings such as public institutions, which is recommended by most researchers [38]. Though the success of the intervention has not yet been evaluated in relation to health outcomes, South Africa sets a good precedent for the continent through the adoption of comprehensive mandatory regulations to limit salt levels permitted in a wide range of processed foods [47]. Furthermore, subsidizing healthy food purchases among health plan members (South Africa’s largest private health insurance company, where members receive up to 25% cash back on healthy food purchases) appears to be a promising intervention to reduce salt intake [48]. To adapt such price discounting strategies to the Ethiopian context, budget allocations would need to cover the cost of the availability of private health insurance companies. The collection of food purchasing data and data on the nutritional composition of foods is also important to support the implementation and monitoring of such strategies.

This study has highlighted the existence of the multi-sectoral food and nutrition governing body to oversee the implementation and monitoring of the relevant policies in Ethiopia, which demonstrates good commitment and an understanding of the importance of systems of accountability, in line with the political declaration of the high-level meeting of the general assembly on the prevention and control of NCDs [49]. However, policy documents directly addressing NCDs are still primarily the responsibility of the Federal Ministry of Health, and there does not appear to be a coordinated approach to addressing diet-related NCDs, including reducing salt, across relevant ministries. The health sector should display more collaborative and distributive leadership to enable effective governance and build leadership capacity across sectors and all levels of government to cultivate champions in different sectors who can agree on common objectives [50]. To further ensure progress towards the achievement of diet-related NCDs, the Ethiopian government could implement the four-step accountability framework developed by Kraak VI et al., 2014 [49].

Our analysis also highlighted the absence of budgetary allocations and established mechanisms for monitoring and evaluating diet-related NCDs. This might be due to the fact that budgets are included in implementation plans and other documents that we were not able to access, but it could also reflect a lack of political commitment. Significant investments are required to promote healthy nutrition and discourage unhealthy diets with a view to achieving SDG 3, “Good Health and Well-Being” [47]. A total of USD 258,539,000 is required from 2020/21 to 2024/25 in order to implement the highest priority interventions and the national level NCD response, which would be estimated to consume 6.1% of the estimated budget [51]. Healthy choices in diet are only possible when good food is available and affordable—so SDG 1, No Poverty, and SDG 2, Zero Hunger, are inseparable from SDG 3. As the world grapples with the economic consequences of the COVID-19 pandemic, health financing at national and global levels risks being squeezed, along with projected increases in global poverty and economic inequality [52]. To meet the SDG 3 targets, fiscal measures, including the use of price and taxation, should be considered.

More than half (*n* = nine) of the food and nutrition-related programs identified through this review focused on addressing undernutrition in the critical period of the first 1000 days between conception and a child’s second birthday. Intervention efforts focused on the first 1000 days were not originally designed as a measure for NCD control. However, it is now included in the NCD Best Buys as there is increasing evidence that prenatal deprivation due to poor nutrition and poverty, among other reasons, can cause irreversible changes in the way that fetal organs and systems develop [46,53]. These changes, in conjunction with inadequate care in the first two years of a child’s life, can significantly increase the risk of developing diabetes, hypertension, bone disease and mental illness later in life [46]. Therefore, global technical and financial support needs to target the first 1000 days as well as other NCD prevention activities as part of the “ONE common agenda” for Ethiopia.

Despite the right to health for every Ethiopian being guaranteed in the 1995 Constitution of the Federal Democratic Republic of Ethiopia [54], this does not appear to translate into specific policy commitments based on considerations of equity and clear articulation of human rights within the policy documents examined. Our analysis also found both a lack of consistent mention of populations likely to be at risk of NCDs or likely to suffer more from their impact and a virtual absence of policies to protect vulnerable populations, including on the basis of gender. This finding is consistent with a study performed in South Africa that shows limited progress in implementing right-to-food policies [54]. Without adequate attention to equity in the design, implementation and monitoring of national guidelines, policies and strategic documents, identifying whether some groups are more likely to be differentially exposed to NCDs in general and higher salt intake in particular due to their socio-economic circumstances is very difficult. Therefore, emphasis should be given to tailoring actions within the national documents to ensure that such groups can benefit equally from the national initiatives. Furthermore, to strengthen national efforts to address the determinants of NCDs, national policies should be rights-based and recognize the obligation of the state to respect, protect and fulfill people’s right to healthy diets, including its responsibility to govern the commercial and social determinants of healthy diets using legislative, regulatory and administrative measures at its disposal [55].

A key strength of this policy analysis is the application of the NCD “Policy Cube”—a framework to present three key axes of a robust policy environment to address and prevent NCDs: comprehensiveness, effectiveness and equity. Nevertheless, the study had several limitations. First, most (14/18) of the identified documents are out of date, so it is not clear whether they are still in operation. Second, there is an element of subjectivity in the researcher’s interpretation of the presence or absence of policy features, particularly in those documents that may have lacked specific information (for example, a lack of budgetary details) to make clear judgments. Third, the whole process depended on internet-based data sources. Finally, since policy content analysis only examines texts, it is inherently reductive. Further relevant information can be gleaned from discussions with stakeholders, which will be undertaken in a different study. Despite the aforementioned weaknesses, our study has helped to identify gaps and increase understanding about where and how policies could be strengthened to address the growing burden of NCDs more effectively in Ethiopia.

## 5. Conclusions

The government of Ethiopia has taken positive steps towards addressing NCDs. Nonetheless, our policy analysis reveals that although most of the policy documents address diet-related WHO “best buys” through promoting healthy nutrition and discouraging unhealthy diets, salt reduction measures recommended in the Best Buys were not adequately addressed. Moreover, the policy documents also lack budget details, monitoring and evaluation, rights-based language and the necessary specificity or commitments to ensure that measures improve health equity. Therefore, there is a need for the NCDs and public health communities to come together to call on the heads of federal ministries to embrace and adopt a more holistic policy framework for the prevention of dietary NCDs that covers the breadth of WHO “Best Buys”, strengthens policy authoritativeness, identifying budgetary means, incorporates systems of accountability, promotes a greater focus on inequities in exposure across population groups and adopts human rights-based approaches. Given the role of external partners in supporting development in the country, we encourage them to provide technical and financial support to the government to formulate and implement evidence-informed NCD policies, which include salt reduction programs.

## Figures and Tables

**Figure 1 nutrients-15-01745-f001:**
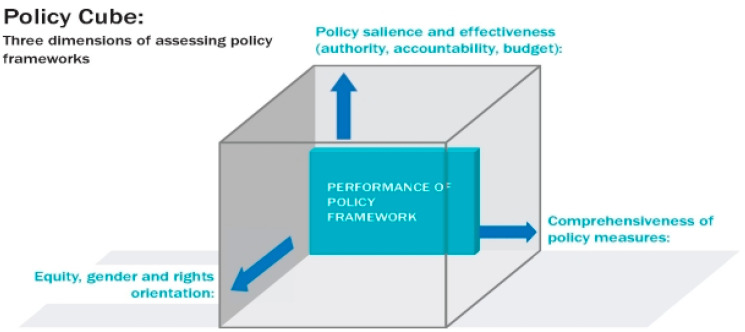
Three dimensions to assess the robustness of diet-related NCD frameworks: the policy cube approach. Reprinted from Buse et al. [26].

**Figure 2 nutrients-15-01745-f002:**
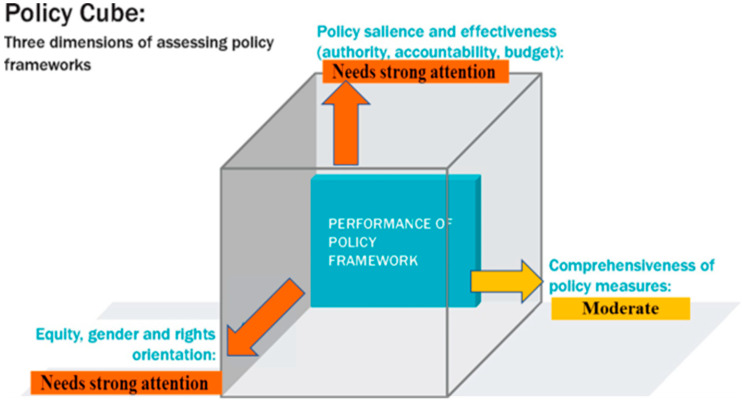
The policy cube: three dimensions of assessing the strength of the salt reduction policy framework in Ethiopia, adapted with permission from Buse et al. [26].

**Table 1 nutrients-15-01745-t001:** List of analyzed documents in this study (number in parenthesis refers to endnotes), 2021 (*n* = 18).

Type of Document	Document Title	Period Covered	Published by	Source (Where Publicly Availiable)
Policy	National Food and Nutrition Policy [21]	2018–2030	Government of Ethiopia	Accessed: 1 January 2023 https://www.nipn.ephi.gov.et/sites/default/files/2020-05/Food%20and%20Nutrition%20Policy.pdf
	Ethiopia National Health Policy [22]	2019–2029	Ministry of Health	Accessed: 1 January 2023 https://www.scribd.com/document/358431814/Ethiopian-National-Health-Policy-PDF#
Proclamation	Food, Medicine and Health Care Administration and Control Proclamation No. 661/2009 [24]	2010–2020	Ethiopian Food and Drug Authority	
Program	National Nutrition Program II [28]	2016–2021	Ministry of Health	Accessed: 1 January 2023 https://faolex.fao.org/docs/pdf/eth190946.pdf
	Second Generation Health Extension Program (HEP) [25]	2016–2026	Ministry of Health	Accessed: 1 January 2023 https://publications.jsi.com/JSIInternet/Inc/Common/_download_pub.cfm?id=16833&lid=3#:~:text=The%20components%20of%20the%20health%20extension%20program%20varies,the%20community%20level%20and%20institutionalizing%20the%20WDA%20platform
	Health sector development program IV [29]	2011–2016	Ministry of Health	Accessed: 1 January 2023 https://leap.unep.org/countries/et/national-legislation/health-sector-development-program-iv-201011-201415#:~:text=This%20Health%20Sector%20Development%20Program%20%28HSDP%29%20IV%20is,Vision%20of%20%22seeing%20healthy%2C%20productive%2C%20and%20prosperous%20Ethiopians%22
	National Nutrition Program—I [30]	2013–2015	Ministry of Health	Accessed: 1 January 2023 https://extranet.who.int/nutrition/gina/sites/default/filesstore/FJI%202014%20SSF%20Strategy.pdf
National Plan or Strategy	National Strategic Plan for the Prevention and Control of Major Non-Communicable Diseases [27]	2021–2025	Ministry of Health	Accessed: 1 January 2023 https://www.researchgate.net/publication/313109049_Guidelines_on_Clinical_and_Programmatic_Management_of_Major_Non_Communicable_Diseases/link/5c29140ba6fdccfc70731577/download
	Nutrition Sensitive Agriculture Strategic Plan [26]	2016–2021	Ministry of Agriculture	Accessed: 1 January 2023 https://faolex.fao.org/docs/pdf/eth174139.pdf
	National Cancer Control Plan [31]	2016–2021	Ministry of Health	Accessed: 1 January 2023 https://apps.who.int/iris/handle/10665/44009#:~:text=Plan%20de%20Accio%CC%81n%202008-2013%20de%20la%20estrategia%20mundial,cardiovascular%20diseases%2C%20cancers%2C%20chronic%20respiratory%20diseases%20and%20diabetes
	National Strategic Action plan (NSAP) for prevention and control of non-communicable diseases in Ethiopia [32]	2014–2016	Ministry of Health	Accessed: 1 January 2023 https://faolex.fao.org/docs/pdf/eth174139.pdf
	The National Health Promotion and Communication Strategy [33]	2016–2020	Ministry of Health	Accessed: 1 January 2023 https://extranet.who.int/nutrition/gina/sites/default/filesstore/FJI%202014%20SSF%20Strategy.pdf
	National school health and nutrition strategy [34]	2013–2018	Ministries of Education, Health and Agriculture	Accessed: 1 January 2023 https://healtheducationresources.unesco.org/library/documents/national-school-health-and-nutrition-strategy
	Agriculture and Natural Resources Sector Growth and Transformation Plan II [35]	2015–2020	Ministry of Agriculture and Natural Resources	Accessed: 1 January 2023 https://www.medbox.org/document/ethiopia-health-sector-transformation-plan-201516-201920-2008-2012-efy#GO
	Agricultural growth program II [36]	2015–2020	Ministry of Agriculture and Natural Resources	Accessed: 1 January 2023 https://www.gafspfund.org/projects/agricultural-growth-project-ii-agp-ii
	Ethiopia Nutrition Advocacy Plan [37]	2013–2018	Ministry of Health	Accessed: 1 January 2023 https://www.fantaproject.org/sites/default/files/resources/Ethiopia-Nutrition-Advocacy-Apr2013.pdf
Guideline	Guidelines on Clinical and Programmatic Management of Major Non-Communicable Diseases [33]	2016–2021	Ministry of Health	Accessed: 1 January 2023 https://www.researchgate.net/publication/313109049_Guidelines_on_Clinical_and_Programmatic_Management_of_Major_Non_Communicable_Diseases/link/5c29140ba6fdccfc70731577/download

**Table 2 nutrients-15-01745-t002:** Characteristics of included documents, 2021 (*n* = 18).

SN	Policy Title, Policy Type, Time Frame, Policy Authority, Reference	NCD Policy Cube Components		
		Comprehensiveness of Policies	Policy Salience and Effectiveness	Equity, Gender and Human Rights Orientation
1	National Food and Nutrition Policy ^a,M^ 2018–2030 [21]	**NCDs and Nutrition**—interventions addressing nutritional problems during pregnancy, infancy, childhood and adolescence to prevent the risks of diet-related problems, chronic non-communicable diseases, disability and mortality by giving great focus to the critical periods of the first 1000 days (pregnancy through the first two years of a child’s life). General statements about a healthy diet; **Salt**—Nil.	**Accountability**—food and nutrition governing body bestowed with the required authorities, accountability, legal framework and functional organizational structure from federal to kebele levels is responsible for facilitating the implementation of food and nutrition activities; **Resources**—mobilize from the Ethiopian Government Treasury, partners, the private sector and community participation. Priority is given to capacity-building efforts; **Monitoring**—link with existing monitoring and information systems and strengthen the capacities of the regulatory agencies at national, regional and local levels for food policy. No specific requirements for NCDs or salt reduction.	**Gender**—acknowledges gender inequality as a cause and effect of all forms of malnutrition. Involvement of women, girls, males, influential community leaders and community participation is given due consideration in addressing gender inequality and improving the nutritional status of women and adolescent girls by breaking the intergenerational cycle of malnutrition; **Equity**—addresses narrowing vulnerability and inequalities about sectoral, gender, disability, age, geography, social status and living styles; **Human rights**—the provision of adequate, safe, high quality and nutritious food is considered a human right of all citizens and ensures the right to an informed choice of foods. Food and nutrition security are stated as a human and constitutional right of all citizens. The provision of all food and nutrition services shall consider the human rights of all citizens.
2	National Nutrition Program II ^c,M^ 2016–2021 [28]	**NCDs and Nutrition**—public awareness on healthy dietary behaviors and prevention of diet-related chronic non-communicable diseases a framework for coordinated implementation of nutrition interventions (including optimal breastfeeding, optimal complementary feeding, mitigation, prevention of micronutrient deficiencies, water sanitation and hygiene, deworming, food fortification and management of acute malnutrition) in order to end hunger by 2030; promotes healthy diets through the importation of fortified food (edible oil, salt), increased consumption of fruits and vegetables, and reduced consumption of soda beverages, saturated fats and trans-fatty acids; Prioritises adolescent, maternal, infant and young child nutrition; **Salt**—Nil.	**Accountability**—a national nutrition coordinating body, under the Office of the Deputy Prime Minister, is responsible and has authority and resources. Overall, 22 departments (including the Ministry of Health, Ministry of Women and Children, Ethiopian Food and Drug Administration, Ministry of Trade, Ministry of Industry, Ministry of Education, and Ministry of Agriculture) have responsibilities; **Resources**—USD 1.1 billion estimated for implementation over 5 years; **Monitoring**—utilize existing monitoring and evaluation activities for NCDs and food policies: strengthening surveillance and health systems research on major NCD risk factors by the Ethiopian Public Health Institute and the Ethiopian Institute of Agricultural Research; enforcing and regulating the activities of manufacturers, importers and distributors of products and supplies; conducting regular inspection and monitoring of food processing factories; registering and issuing market authorization for nutritious food products and developing a marketing strategy for fortified foods.	**Gender**—promotes the integration of gender into sectoral nutrition implementation programs, strategies and guidelines; **Equity and human rights**—nil.
3	Health Sector Transformation Plan ^d,M^ 2016–2021 [39]	**NCDs**—general statements on NCD prevention and control, including addressing unhealthy diets and alcohol consumption; **Nutrition**—recognized as a cross-cutting issue that contributes to progress towards several millennium development goals; **Salt**—no mention of salt reduction. Highlights the importance of the availability of quality-assured iodized salt.	**Accountability**—the Ministry of Health; **Resources**—Financial and human resources are allocated: USD 14.1 billion; **Monitoring**—nutritional indicators are clearly set for the prevention and control of NCDs and food/nutrition policies, but not for salt reduction.	**Equity**—considerations for equitable access to essential health services to reduce disparities between regions and groups with different levels of underlying social advantages/disadvantages; **Gender**—creating equal opportunities and bringing health differentials down to the lowest possible level through strengthening gender mainstreaming at all levels of the health care system; **Human rights**—nil.
4	Guidelines on Clinical and Programmatic Management of Major NCDs ^e,L^ 2016–2021 [33]	**Nutrition**—healthy nutrition (adequate consumption of fruits, vegetables, healthy and traditionally acceptable foods) is encouraged, and unhealthy diets with excess sodium (during processing and cooking), sugar, saturated fat and trans-fat are discouraged to reduce the burden of NCDs; Nutrition policy and legislation recommended: ○ Subsidies on vegetables and fruits; ○ Increasing taxes on processed foods and drinks to decrease consumption of excess salt, sugars, saturated fats or trans fatty acids and/or refined carbohydrates; ○ Packaging and labeling of factory-processed foods and drinks, including detailed nutritional content; ○ Regulation of promotion and advertisement of processed foods and alcoholic beverages. **Salt**—diets high in salt are discouraged. Taxes on processed foods (e.g., high-salt foods) are mentioned.	**Accountability**—offices at different levels of the health sector, from the Federal Ministry of Health to Regional Health Bureaus (policy matters and technical support) and Woreda Health Offices (managing and coordinating the operation of a district health system); **Resources**—there is a budget to cover the health workforce, financial resources, infrastructure, equipment and medicines; **Monitoring**—key indicators for the major NCDs program are monitored by collecting data at regular intervals and continuous evaluation of the processes and outcomes of the national NCDs is in place.	**Equity**—promotes equitable NCD prevention and control services in primary care; **Gender and Human Rights**—nil.
5	National Cancer Control Plan ^d,M^ 2016–2021 [31]	**NCDs and Nutrition**—healthy diets (diets high in fruits and vegetables) are an important approach to cancer control. Nutritional support services for cancer patients through community and home-based palliative care should be strengthened. The plan recommends the government introduces control of the import of processed foods having high fat, sugar and salt.	**Accountability**—offices at different levels of the health sector, from the Federal Ministry of Health to Regional Health Bureaus and Woreda Health Offices; **Resources**—there is a budget to cover the health workforce, financial resources, infrastructure, equipment and medicines; **Monitoring**—enforcement of the importation of processed foods having high fat, sugar and salt and monitoring indicators for cancer control are clearly set.	**Equity**—equitable accessibility of services is a guiding principle of the National Cancer Control Plan; **Gender and Human Rights**—nil.
6	Second Generation Health Extension Program (HEP) ^c,M^ 2016–2026 [25]	**NCDs**—includes NCD prevention through strengthening the provision of primary health care by qualified community health workers, upgraded health post infrastructure and an expanded package of community health services; **Nutrition**—behavioral interventions promoting healthy nutrition (adequate consumption of fruits, vegetables, and healthy and traditionally acceptable foods) and discourages unhealthy diets with excess sodium, sugar, saturated fats, and trans-fat; **Salt**—recommends dietary changes by restricting salt, including reducing salt when cooking and limiting processed and fast foods.	**Accountability**—the Ministry of Health is accountable for implementation and monitoring, but the community health workers do the implementation work; **Resources**—no specific budget was mentioned, but it highlights budget needs to be allocated for the health workforce, financial resources, infrastructure, equipment and medicines; **Monitoring**—there needs to be control over the import of processed foods having high fat, sugar and salt, and monitoring indicators for cancer control are clearly set.	**Gender, Equity and Human Rights**—nil.
7	National Strategic Plan for the Prevention and Control of Major Non-Communicable Diseases 2020/21–2024/25 [27]	**NCDs**—nil; **Nutrition**—increase public awareness of a healthy diet including adequate intake of vegetables, fruits and whole grains; limited intake of sugar, salt and saturated fat and avoidance of trans-fats. Target to reduce insufficient fruit and vegetable consumption in persons aged 15+ years by 2025 (a 25% relative reduction). Liaise with the Ministry of Education to introduce healthy diet promotion in school curriculums and awareness-creation activities; **Salt**—reduce the mean population’s salt intake to <5 g/d in persons aged 15+ years by 2025 (30% relative reduction): ○ Supportive environment: reduce salt intake through the establishment of a supportive environment in public institutions such as hospitals, schools, workplaces and nursing homes to enable lower sodium options to be provided; ○ Educate: reduce salt intake through behavior change communication and a mass media campaign. ○ Packaging—reduce salt intake through the implementation of front-of-pack labeling. Reformulate food—reduce salt intake through the reformulation of food products to contain less salt and the setting of target levels for the amount of salt in foods and meals.	**Accountability**—national steering committee for NCDs comprising all relevant sectoral ministries (e.g., Ministy of Health, Education, Trade, Agriculture and Finance), as well as non-governmental actors (e.g., UN agencies, civil society organizations) The MOH-Ethiopia is responsible for setting standards, developing and revising national guidelines, preparing national action plans, monitoring and evaluation, advocacy and operational research and overseeing the overall national coordination of health services and programs for NCDs and risk factors. RHBs take the technical guidance from the MOH and adopt it per their regional context to implement interventions on the prevention and control of NCDs and risk factors. The RHBs are in charge of planning, coordinating, implementing, monitoring and evaluating the health sector response to NCDs and risk factors in their respective regions; **Resources**—USD 258,539,000 over 5 years and successful implementation of the program requires timely mobilization of resources **Monitoring**—Existing monitoring and evaluation activities for NCDs are outlined in the document, including incorporating key indicators of NCDs into the national health information Management System (HMIS).	**Gender**—nil; **Human rights and equity-based**—empowerment of individuals and communities were mentioned as guiding principles of the strategic plan.
8	National Strategic Action plan (NSAP) for prevention and control of non-communicable diseases in Ethiopia ^d,M^ 2014–2016 [32]	**NCDS**—nil; **Nutriton**—a healthy diet including culturally appropriate, affordable and balanced dietary habits (proper intake of vegetables, fruits, and whole grains; low-fat dairy products, poultry, fish, legumes, vegetable oils and nuts; and limited intake of sweets, sugar-sweetened beverages and meats) is encouraged. Unhealthy diets with excess sodium, sugar, saturated fat, and trans-fat for individuals and the population at large; **Salt**—public policies protecting people from excessive salt consumption are cited as a WHO “best buys” intervention through the establishment of a supportive environment in public institutions.	**Accountability**—The Federal Ministry of Health NCD team and Regional Health Bureau, Woreda Health Office, health facilities, development partners and civic society are responsible for implementing the program; **Resource**—funding is allocated to support a trained health workforce, infrastructure, essential medicines, diagnostic, palliative and therapeutic technologies; **Monitoring**—processes for monitoring outcomes, specifically morbidity and mortality by disease; behavioral risk factors; and assessing health system capacity to respond to the population’s NCD-related health needs; enforcing regulations to control local production of fruits, vegetables and other foods that contribute to a healthy diet; and import marketing of processed food and drink that contain high levels of added salt, sweeteners, saturated, trans-fatty acids and refined carbohydrates.	**Gender and Equity**—ensuring equity and promoting gender equality are priorities; **Human rights**—nil.
9	Ethiopia Nutrition Advocacy Plan ^d,L^ 2013–2018 [37]	**NCDS**—nil; **Nutrition**—malnutrition. Effective nutrition interventions to improve nutritional status through support and promotion of breastfeeding food fortification, locally produced specialized food products and salt iodization; **Salt**—nil.	**Accountability**—offices at different levels of the health sector, from the Federal Ministry of Health to Regional Health Bureaus and Woreda Health Offices, are responsible for implementation; **Resources**—funding is allocated towards the health workforce, financial resources, infrastructure, equipment and medicines; **Monitoring**—mechanisms are related to establishing quality assurance systems to monitor the iodine content of iodized salt.	**Gender**—gender issues are mainstream in all health planning and monitoring and evaluation processes; **Equity**—nil; **Human rights**—nil.
10	National Nutrition Program—I ^c,M^ 2013–2015 [30]	**NCDs**—a general statement about the link between NCD risk factors (physical inactivity and an unhealthy diet) and diseases like cardiovascular disease, diabetes, cancer, hypertension and stroke; **Nutrition**—promotes increased consumption of fruits and vegetables and avoids unhealthy behaviors. **Salt**—nil;	**Accountability**—the National Nutrition Coordinating Body, under the Office of the Deputy Prime Minister; **Resources**—financial resources are allocated: USD 547 million; **Monitoring**—indicators for nutritional improvement of adolescent girls, pregnant women, infants 0–6 months old, and infants and young children 6–24 months old are set out.	**Gender**—recognizing gender inequalities can be both a cause and an effect of hunger and malnutrition, gender should be considered in the implementation of nutrition interventions; **Equity**—nil; **Human rights**—nil.
11	Ethiopia National Health Policy ^a,M^ 2019–2029 [22]	**NCDs**—a general statement about dietary habits and major non-communicable diseases and their risk factors; **Nutrition**—focuses on implementation of interventions to prevent health problems caused by food and nutrition; **Salt**—nil.	**Accountability**—the Ministry of Health; **Resources**—mentions financial, technological and human resource allocation but no specific information **Monitoring**—nil.	**Gender**—provision of necessary support to empower women in decision-making roles and in the health of themselves, their families, their communities and their environment is another focus area; **Equity**—ensuring equity and access so all people, particularly the poor and vulnerable, are entitled to have equitable access to quality health services; **Human rights**—nil.
12	Health sector development program IV ^c,M^ 2011–2016 [29]	**NCDs**—a general statement about the need to address major NCD risk factors (alcohol, smoking, diet and exercise) through the health extension program; **Nutrition**—coordinated implementation of nutrition interventions (includng optimal breastfeeding; optimal complementary feeding; mitigation and prevention of micronutrient deficiencies such as vitamin A, iron, and iodine, WASH, deworming; food fortification and management of acute malnutrition); **Salt**—only mentions the importance of utilizing iodized salt.	**Accountability**—offices at different levels of the health sector, from the Federal Ministry of Health to Regional Health Bureaus and Woreda Health Offices, are responsible for implementation; **Resources**—funding is allocated towards human resources, tools and equipment; **Monitoring**—only integrated supportive supervision is utilized for this policy as a means of monitoring and evaluating NCDs programs and targets for the prevention and control of NCDs.	**Gender**—gender equality and empowerment of women; **Equity**—nil; **Human rights**—nil.
13	Food, Medicine and Health Care Administration and Control Proclamation No. 661/2009 ^b,H^ 2010–2020 [24]	**NCDs**—nil; **Nutrition**—standards for nutrition labeling, licensing and regulating trans-regional food safety and quality. Mandates of producers or distributors of salt for human consumption to ensure it meets the standard requirement for iodine content; **Salt**—nil.	**Accountability**—the Ethiopian Food and Drug Administration is accountable for implementation and monitoring; **Resources**—nil; **Monitoring**—nil.	**Gender**—nil; **Equity**—nil; **Human rights**—nil.
14	The National Health Promotion and Communication Strategy ^c,M^ 2016–2020 [33]	**NCDs**—a general statement about the prevention and control of NCDs and creating an enabling environment to promote lifestyle practices (by tackling smoking, alcohol, physical inactivity, processed foods with added salt, sugar and saturated fat, etc.); **Nutrition**—promotion of early and exclusive breastfeeding; **Salt**—nil.	**Accountability**—FMOH and its structural offices at all levels are accountable for the implementation of the strategy. **Resources**—nil; **Monitoring**—continuous monitoring, evaluation, and dissemination of best practices at different levels in collaboration with key stakeholders.	**Gender**—a specific focus is given to gender sensitivity and adolescent needs. The strategy also ensures inclusiveness by gender, age, occupational status, etc.; **Equity**—nil; **Human rights**—nil.
15	National school health and nutrition strategy ^c,M^ 2013–2018 [34]	**NCDs**—nil; **Nutrition**—school feeding programs—providing balanced meals for children in schools, especially those coming from poor and food insecure households and areas, as well as those affected by natural and man-made emergency situations. Micronutrient deficiencies such as vitamin A, iodine and iron, among others, which directly or indirectly affect cognition and can result in better school performance; **Salt**—nil.	**Accountability**—offices at different levels of the Ministries of Education, Health and Agriculture and other relevant ministries are responsible for implementation; **Resources**—nil; **Monitoring**—recommends a strong monitoring and evaluation system to achieve health and educational outcomes.	**Gender**—programs should be sensitive to the different needs of boys and girls; **Equity**—in response to the specific needs of girls, children with disabilities, orphans and other vulnerable children; **Human Rights**—every child has a right to quality education, quality health and nutrition services, privacy and confidentiality regarding their health.
16	Nutrition Sensitive Agriculture Strategic Plan ^c,M^ 2016–2021 [26]	**NCDs**—nil; **Nutrition**—it acknowledges diversifying diet as being pivotal to the attainment of food and nutrition security. Agricultural development that puts nutritionally rich foods, dietary diversity, and biofortification at the heart of overcoming malnutrition and micronutrient deficiencies; **Salt**—nil.	**Accountabiltiy**—the Food and Nutrition Unit of MOA will be primarily responsible for coordinating the mainstreaming of nutrition in the agriculture sector at all levels; **Resources**—nil; **Monitoring**—the plan promotes a comprehensive monitoring and follow-up mechanism for multi-sector NNP implementation activities.	**Gender**—ensuring gender equality in nutrition-sensitive agricultural intervention programs through empowering women and enhancing their role in nutrition-sensitive agriculture; **Equity**—nil; **Human rights**—nil.
17	Agriculture and Natural Resources Sector Growth and Transformation Plan II ^c,M^ 2015–2020 [35]	**NCDs**—nil; **Nutrition**—nutrition dense products to be achieved through interventions mainstreamed across other program areas. **Salt**—Nil.	**Accountability**—the Federal Ministry of Agriculture and Natural Resources is mandated to take the leadership role to ensure the preparation, implementation, monitoring and evaluation of the plan at all levels and will ensure responsibility and accountability by the different sectors; **Resources**—budget needed to implement the set strategic goals is clearly stated (ETB 41,154,934,000); **Monitoring**—a strong and gender-sensitive monitoring system is a crucial tool to help the stakeholders know whether the implementation has yielded the desired result for the development of the country.	**Gender**—make gender equity mainstream; **Equity**—considers the needs of women, farmers, youth, pastoralists, agro-pastoralists and food-insecure populations; **Human rights**—nil.
18	Agricultural growth program II (AGP-II) ^c,M^ 2015–2020 [36]	**NCDs**—nil; **Nutrition**—crop and animal productivity to improve the consumption of diversified diets at the household level through: (i) diversification of crop, fruit and livestock production; (ii) promotion of appropriate technologies for food production and processing through the handling, preparation and preservation of food, supporting nutritious food utilization and (iii) supporting local complementary food production; **Salt**—Nil.	**Accountability**—steering committee at the federal, regional and woreda levels will be accountable; **Resources**—two major sources: IDA and USAID/UNDP. Implemented through the Ministry of Agriculture; **Monitoring**—monitoring and evaluation system linked with other existing monitoring and information.	**Gender**—strengthening the capacity of women and men to provide for the food security, health and nutrition of their families, increase access to year-round availability of high-nutrient content food and improve nutrition knowledge among rural households to enhance dietary diversity; **Equity**—special focus on women and youth groups, resource-poor farmers and large segments of rural poor communities and urban and per-urban areas; **Human rights**—nil.

Abbreviations for the document type: Policy = ^a^; Proclamation = ^b^; Program = ^c^; National plan = ^d^; Guidelines = ^e^; Policy authority: High = ^H^, Moderate = ^M^, Low = ^L^.

**Table 3 nutrients-15-01745-t003:** Hierarchy of the authoritativeness of policy documents, 2021 (*n* = 18).

Hierarchy	Type of Document
Highest level of authority	Food, Medicine and Health Care Administration and Control Proclamation No. 661/2009 [24]
Moderate level of authority	Nutrition Sensitive agriculture Strategic Plan [26]
	The National Health Promotion and Communication Strategy [33]
	National school health and nutrition strategy [34]
	Agriculture and Natural Resources Sector Growth and Transformation Plan II [35]
	Agricultural growth program II [36]
	National Nutrition Program II [31]
	Second Generation HEP [25]
	National Nutrition Program I [30]
	Health Sector Development Program IV [29]
	National food and nutrition policy
	Ethiopia National Health Policy [22]
	Health Sector Transformation Plan [39]
	National cancer control plan [32]
	NSAP for prevention and control of non-communicable diseases in Ethiopia [32]
	National Strategic Plan for the Prevention and Control of MajorNon-Communicable Diseases [27]
Lowest level of authority	Guidelines on Clinical and Programmatic Management of Major Non-Communicable Diseases [33]
	Ethiopia Nutrition Advocacy Plan [37]

## Data Availability

The materials described in the manuscript, including all relevant raw data, will be available from the corresponding author on reasonable request.

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
