# Peer review of "Do Food and Nutrition Policies in Ethiopia Support the Prevention of Non-Communicable Diseases through Population-Level Salt Reduction Measures? A Policy Content Analysis"

_nutrients, 2023, doi:10.3390/nu15071745_

Round 1
Reviewer 1 Report
This is important work to demonstrate the progress and potential outcomes and impact of policies to promote healthy food environments and prevent NCD in Ethiopia through sal reduction measures and may contribute to the implementation of new policies or the reformulation/update of current policies.
I have only two minor questions for the authors:
- Did the implementation of comprehensive public policies in the mentioned documents consider the marketing of foods high in salt in order to reduce its impact?
- Has the implementation of policies related to the promotion of a healthy food offer in public spaces been considered, in order to promote healthy food choices, as well as to encourage and support private sector companies to adopt similar policies?
Reviewer 2 Report
Dejen Tekle and colleagues showed us the work entitled "Do food and nutrition policies in Ethiopia support the prevention of non-communicable diseases through population-level salt reduction measures? A political content analysis".
The focus of the paper is on the use of sodium in the diet, but without comprehensive implications.
The availability of health care for such a poor country is unknown, as is the possible application of antihypertensive therapy if the population has hypertension.
It is impossible that sodium is the cause of death in newborns...
there is no information about which foods are rich in sodium, and how it can be converted just by recommending to take fruits and vegetables.
There is no conclusion! There are no facts which are concrete proposals to reduce the use of table salt in the diet.
Reviewer 3 Report
General Comment
- The article is very pertinent for the design of public health policies, specifically in the area of excessive salt consumption.
- Some bibliographic references that support the text are very old, so the scientific evidence in this area should be updated (e.g. ref 18, ref 19)
- Some issues that can be improved related to the methodology.
- Line 54 - "The World Health Organization", should be "the World Health Organization"
Methodology
- Lines 139-142. The terms used to search were in english? The oficial language in Ethiopia is the english? The original terms used in the search should be clearly mentioned in the paper, at least in appendix.
- Line 146 - the paper published between 2010 and 2022 were included? The abstract refers inclusion until december 2021. This should be corrected.
- The methodology should be improved in the description of selection, summarizing and reporting the results. The selection was made by one or two independent researchers? If the two researchers have different decisions, what was the criteria for final decision?
Results
- The 33 final policy documents can be summarized in one appendix to the paper?
- Table 2 the results described can be more focused on salt reduction measures, in order to make the table more succinct and atractive.
Round 2
Reviewer 2 Report
/